# Auditive beta stimulation as a countermeasure against driver fatigue

**Michèle Moessinger[1]☯, Ralf Stürmer[2]☯, Markus Mühlensiep [3]☯ ***

**1** Groupe Renault, Direction de la Recherche / User Experience, Guyancourt, France, **2** FOM Hochschule, Wuppertal, Germany, **3** Infrasonics GmbH, Köln, Deutschland

☯ These authors contributed equally to this work.
* m.m@infrasonics.de

**Data Availability Statement:** All relevant data are within the paper and its Supporting Information files.

**Funding:** This research was supported by the Groupe Renault1 (https://group.renault.com) and Infrasonics GmbH2 (https://www.infrasonics.de).

## Abstract

### Objective

Fatigue in road traffic can occur after long driving times or during night-time driving and can lead to a dangerous decrease in vigilance or microsleep. As a countermeasure, the effectiveness of brain stimulation by means of frequency-modified music is investigated. Small frequency shifts between two different sounds simultaneously perceived by both ears (e.g. 400 and 418 Hz) stimulate the brain to increase activity in the stimulated range (e.g. 18 Hz).

### Methods

The effects of acoustic brain entrainment (ABE) in the EEG beta range (12–20 Hz) were compared to placebo during day and night driving (n = 40 each) in 80 subjects. The effects were examined at the subjective (Karolinska Sleepiness Scale, KSS), physiological (EEG) and performance level (test battery for attention testing, TAP). The test drive took place on the motorway. Sequence of events with measurement times: Preparation in the laboratory (60 min; TAP & KSS), driving (90 min day/ 180 min night; KSS every 30 min), beta/placebo stimulation while driving (20 min), rest (20 min; TAP & KSS), driving (60 min; KSS every 30 min), end of driving (40 min; TAP & KSS). EEG was recorded continuously (Fz, Cz, Pz, C3, C4) and analyzed for 10 min time windows.

### Results

The subjective fatigue rating in the KSS decreased significantly after ABE compared to placebo. This was still significant about 100 minutes after stimulation. The ABE led to a significant increase in EEG beta activity compared to placebo. This was still significant about 80 minutes after stimulation. Furthermore, the ABE led to a significant decrease in theta activity compared to placebo. This was still significant about 70 minutes after stimulation and was more pronounced during daytime driving. Faster reaction times were observed for the ABE compared to the placebo condition during day and night driving. The faster reaction times were partly significant for the ABE still 80 minutes after stimulation.

The support was provided in the form of salary1 (MM, RS, MM) and provision of research equipment1&2. The funders had no role in study design, data collection and analysis, decision to publish, or preparation of the manuscript.

**Competing interests:** The authors have read the journal's policy and have the following conflicts: This research was supported by the Groupe Renault and Infrasonics GmbH. The authors have no further relevant declarations relating to employment, consultancy, patents, products in developments or marketed products. These sources of funding do not alter the authors' adherence to all the PLoS ONE policies on sharing data and materials.

## Conclusion

Positive effects of ABE during driving could be demonstrated on the subjective, physiological and performance level. The effect was more pronounced during the day than at night. No negative side effects of Beta Stimulation were observed. The stimulation in the beta frequency range led to an increase of beta activity in the EEG.

## Introduction

Driver Fatigue is a serious problem on roads all around the globe. Long journeys, sleep deprivation and monotonous environment can result in dangerously decreased vigilance or even microsleep episodes. Accordingly, driver tiredness or fatigue is commonly regarded as a major influencing factor for traffic accidents [1, 2]. Maia et al. [3] found that drowsy driving seems to be a fairly common phenomenon, particularly in self-identified short-sleepers and insufficient sleepers. Pylkkönen et al. [4] provided evidence, that driver education alone, by means of an alertness management training, is not a sufficient countermeasure as such to alleviate driver sleepiness. Driver fatigue has been extensively studied for many years, and beside its influence on self-rated tiredness and performance measures, its psychophysiological assessment, with indicators derived from the electroencephalogram (EEG) as key measures to quantify sleepiness, plays an important role (e.g. [5, 6]).

The neurochemical processes in the brain take place in certain wave patterns. These wave patterns can be made visible by measuring the changes in electrical voltage at the surface of the scalp. This measurement and representation is called electroencephalogram (EEG). The EEG commonly distinguishes five different patterns, which are determined by the predominant frequency that have a relation to human arousal states (compare for example [7]). Generally, higher frequency EEG activity is related to higher arousal levels [5, 8, 9].

1. high beta waves: Frequency 22 - <35 Hz, can be observed at high concentration levels.

2. low beta waves: Frequency 13 - <22 Hz, occur mainly in the awake state but also in REM sleep, related to cognitive activity, alertness

3. alpha waves: Frequency 8–12 Hz, found in a relaxed awake state, especially when the eyes are closed, relaxation

4. theta waves: Frequency 4–8 Hz, occur in light sleep, increased during drowsiness and inattention

5. delta waves: Frequency < 4 Hz, are characteristic for deep sleep.

A theoretical background for arousal state related human performance decrements will be provided in the following section. Furthermore, the background for a specific acoustic stimulation technique will be outlined as a fatigue and tiredness countermeasure.

### Key concepts: Fatigue, tiredness, sleepiness and sleep

Fatigue, tiredness, and sleepiness are terms to describe the transitory period between awake and sleep [5] and to explain the likelihood of falling asleep on one hand and the influences on performance on the other hand. To explain the human need for sleep, two different kinds of theories have been proposed: 1) recuperation theories and 2) circadian theories. An integrative combination of recuperation and circadian theories is outlined by the "Two-Process-Model" [10]. Thus, the "Two-Process-Model" specifies the combined impact of a homeostatic need for

sleep, increasing with hours of wakefulness, and circadian rhythms driven by an internal clock. The fundamental changes in neural functions involved during the regulation of sleep and wakefulness have been described for example by Saper, Scammell, and Lu [11]. Tiredness and sleepiness can be defined as a psychophysiological arousal state influenced by sleep deprivation, sleep quality and circadian rhythms.

Fatigue is the most common used term when it comes to the description and explanation of the temporal-limited human operator performance decrements in the relevant literature. Fatigue can be classified into physical and mental categories. The focus of the current article will be on concepts related to mental categories of fatigue that has been defined as the likelihood of falling asleep. Mental fatigue is most often conceptualized as being task related [12] and can occur either due to high workload (active task related fatigue) or due to monotony (passive task related fatigue).

Saxby et al. (2013) pointed out that the different patterns of fatigue should be taken into consideration when developing and using automated driving devices and that passive fatigue (due to monotony) poses a more immediate threat to safety than active fatigue [13]. In conclusion, as applied to the road environment, fatigue is seen as a general psychophysiological state which diminishes the ability to perform the driving task by altering alertness and vigilance.

Given the background that fatigue and high levels of tiredness are major influencing factors for traffic accidents, countermeasures in this area can play a vital role in enhancing road safety. The technique and neuropsychological background of ABE is outlined in the next section.

## Acoustic Brain Entrainment (ABE)

Acoustic brain entrainment describes the mechanism of the brain to synchronize its brainwave frequencies with the rhythm of external periodic auditory stimulation. Because the frequency of brain waves as reflected in EEG activity is systematically related to human arousal states, an obvious application scenario is the use of acoustic stimulation to reduce tiredness-related limitations of driving abilities.

**Auditory beats.** The brain can be trained to produce specific EEG frequencies by external stimulation. With visual stimulation, flashes of light are presented at the desired frequency (e.g. for alpha state 8 to 13 cycles per second). In the auditory range this is more complicated: The desired frequencies are partly not perceivable by the human ear, as human hearing is limited to approximately 20–20000 Hz. Here, the phenomenon of binaural and monaural auditory beats is used. When two tones with slightly different frequencies are presented separately to the left and right ears the listener perceives a single tone that varies in amplitude at a frequency equal to the frequency difference ($\Delta f$) between the two tones [14]. Binaural beats are an auditory phenomenon that has been suggested to alter physiological and cognitive processes including vigilance (e. g. [15]). Binaural beat sensations depend upon a combination of two different temporally encoded tones in the Central Nervous System (CNS), most likely originating in the superior olivary nuclei. For example, a presentation of a 500 Hz tone to one ear and a 520 Hz tone to the other one, results in a 20 Hz tone perception. On the other hand, monaural stimulation means that the sum of a 500 Hz and a 520 Hz tone is presented to both ears simultaneously. This leads to an interaction at the cochlear level and also results in a perception of a 20 Hz tone.

The discovery of the phenomenon of the so-called "3rd notes" was made independently by the Italian violinist Giuseppe Tartini, the German organist G. Sorge and Jean-Baptiste Romieu in the 18th century [16].

These tones, which are therefore only produced in the brain, were first described by the German physicist H.W. Dove in 1839 and are generally known as "binaural beats"(BB). Beats which are also perceived with one ear, are accordingly called "monaural beats"(MB).

Certain characteristics apply to these beats: The frequency of the pulsation corresponds to the difference between the two fundamental frequencies. Following Dove's discovery, further characteristics of the binaural beats that Oster summarized in 1973 [14].

Peterson investigated the characteristics of binaural beats in comparison to monaural beats [17]. Monaural beats are characterized as more defined and easier to count. Binaural beats, on the other hand, tend to change direction ("shift") and are harder to locate and count. When the frequency increases, the impression of left-right movement becomes a circular motion. The higher the frequency, the closer the sound is to that of the monaural beats. Licklider examined the area in which binaural beats can be perceived [18]. He recognized that the upper perception limit for binaural beats depends on the frequency of the fundamental tones. With base tones from 300 to 600 Hz, his subjects could perceive binaural beats up to a frequency of 30 Hz (i.e. the base tones differed by up to 30 Hz). Groen determined in 1964 that the 90–800 Hz range was the range for the base frequencies in which binaural beats could best be perceived [19]. In the range of 850–1100 Hz they were less perceptible. From 1100–1400 Hz the perceptibility dropped strongly, from basic frequencies around 1500 Hz no more binaural beats could be perceived. In contrast, McFadden and Pasanen were able to detect binaural beats of up to 75 Hz in base tones around 3550 Hz [20]. Monaural Beats are subject to much wider frequency limits, i.e. the basic frequencies are more variable.

This frequency response of the brain could be shown in different studies in different designs and for different stimulus material. For example, Karino et al. were able to show that in the temporal cortex in particular, when stimulated at a certain frequency, a clear peak at the corresponding frequency can be detected in a spectral analysis of the MEG [21]. Kuwada et al. observed that the response of most neurons is coupled to the auditory beat stimulus (i.e. 1 Hz frequency difference = 1 Hz response) [22]. When the frequencies of the binaural beat in the ears are reversed, the pattern of interaural phase difference changes simultaneously. The rate of phase change can be changed by higher frequency differences. Some cells have responded either only to binaural beats or only to time delays.

Pratt et al. were also able to detect frequency follow reactions in a more recent study [23]: The response to binaural beats was higher for 250Hz base frequency than for 1000Hz, for 3Hz auditory beats higher than for 6Hz. The most distinct reactions were found in the left temporal lobe. Schwarz and Taylor investigated the difference between monaural and binaural beats [24]. At a base frequency of 400 Hz and a frequency difference of 40 Hz, stimulation with monaural beats produced a brain response at the corresponding frequency (40 Hz) that outlasted stimulation for about 200ms. Binaural beats also caused this reaction, but to a smaller extent. Draganova et al. compared the generation of Auditory Steady State Response (ASSR) for binaural and monaural tones [25]. ASSR were found in the averaged 151-channel whole-head magnetoencephalographic recordings under both stimulus conditions. The authors describe, that, although the binaural and monaural beat interacted at different levels of the auditory system, the initial responses were projected along the afferent auditory pathway and activated common cortical sources.

Lane et al. compared the effects of binaural auditory beats in the EEG beta and theta frequency ranges and its relations to performance within a vigilance task [15]. Presentation of beta-frequency binaural beats yielded to more correct target detections and fewer false alarms than presentation of theta-frequency binaural beats. These results suggest that the presentation of binaural auditory beats can affect psychomotor performance. The authors conclude, that this technology may provide applications for the control of attention and arousal as well as the enhancement of human performance. Also, Beauchene et al. demonstrated, that listening to 15 Hz binaural beats during a working memory task increased the individual participant's accuracy and modulated the cortical frequency response [26].

Recently, Garcia-Argibay et al. conducted a meta-analysis of experimental studies on the influence of auditory beats on various psychological concepts (memory, attention, anxiety and pain perception) [27]. The analysis included 22 studies that met the authors' requirements (including sufficient data to calculate effect sizes). On average, there were medium significant effects (g = .45), which were higher in the anxiety and analgesia categories than in the cognitive concepts (memory and attention).

On the other hand, there are also examples for studies, that could not substantiate binaural stimulation effects [28–30]. It can be stated in conclusion, that evidence from basic research exists for the general effects of monaural and binaural beats, that makes it seem worth to evaluate the effects within a more applied automotive context.

The auditory beats technology used in this study was originally used for sleep therapy, the effectiveness of the method was shown in various placebo controlled studies, showing its effectiveness by means of sleep diaries, EEG studies and polysomnographies [31].

More recently, the auditory brain entrainment used in this study has been scientifically tested with young athletes. A placebo-controlled study of the German Sports University found that participants in the experimental group go to bed earlier and sleep longer [32]. Towards the end of the eight-week study phase, they showed lower sleep latency. In a sleep questionnaire, the sleep quality of the experimental group significantly higher was compared to the baseline measurement.

## Pilot studies

Before the actual experiment, a series of pilot studies was conducted. Two general demands should be matched: ABE should show an objective effect compared to a placebo condition and effects of ABE should be obtained on the physiological level (by means of EEG measurements), the subjective level (by means of tiredness self-ratings) and the behavioral level (by means of response times in standardized response time tasks).

In pilot study 1, the effects of ABE Power Nap and Beta Stimulation were analyzed on a global EEG indicator (Median Frequency) and on self-rated tiredness using the Karolinska Sleepiness Scale [33] right after and 30 minutes after the stimulation periods. The consecutive pilot study 2 was designed based on findings of the preceding study 1. A more fine-grained scale was used for the tiredness self-ratings and analysis of EEG activity was differentiated in the theta, alpha, and beta frequency range. Self-rated tiredness was assessed and performance testing by means of two different response time tasks was added. The final pilot study 3 analyzed the influence of ABE beta and alpha stimulation on EEG beta and alpha activity compared to acoustic placebo stimulation.

Pilot studies 1 and 2 compared the stimulating effect of a regular power nap as a placebo to ABE with Beta Stimulation during driving and a power nap with ABE stimulation. Subjects drove a german highway by night for approximately 4 hours: Highway drive (180 minutes), Experimental Condition 1, 2, or 3 (20 minutes), Highway drive (40 minutes).

Pilot study 3 compared ABE in the alpha (8–12 Hz)- and low beta (13–21 hz) range to music without auditory beats on a one-hour day drive.

In pilot study 1, ABE Beta Stimulation resulted in a significant increase of EEG Median Frequency after the stimulation. This effect was still found 30 minutes after the stimulation period to be partially significant (p < .1). Further, ABE Beta Stimulation resulted in a significant increase of EEG Low beta activity compared to an acoustic placebo stimulation in pilot study 3. An increase of EEG Low beta activity during ABE Beta Stimulation could only be observed on a descriptive level, but was not statistically significant in pilot study 2. The sample size for the ABE Beta Stimulation condition in pilot study 2 was the smallest of all three studies

(N = 8). It cannot be ruled out, that the lack of significance is due to limited test power, in view of the small sample size in this case. In conclusion, some evidence could be obtained (in two out of three studies) for a general increase of cortical arousal caused by the ABE Beta Stimulation, but further research is necessary to clarify the stability of this effect. Self-rated tiredness, however, decreased significantly after ABE Beta Stimulation in pilot study 1 (30 minutes after stimulation) and pilot study 2 (right after and 20 minutes after stimulation). No significant effect for ABE Beta Stimulation on response times could be obtained, although response times decreased on a descriptive level after ABE Beta Stimulation. Further, Power Nap supported by ABE resulted in a significant increase of EEG alpha activity during stimulation in pilot study 2. This increase was significantly higher compared to Power Nap without ABE.

The differences in p-values between pilot study 1 and pilot study 2 may be related to the different sample sizes for the ABE Power Nap condition (pilot study 1: N = 7 vs. pilot study 2: N = 15). Finally, ABE Power Nap significantly decreased response times, as could be demonstrated in pilot study 2, again outperforming the effect of Power Nap without ABE on a statistically significant level. In sum, relative consistent evidence could be obtained from two studies, that ABE enhances the regeneration effect of a power nap on the physiological, subjective and performance level.

In pilot study 3, ABE alpha Stimulation during driving resulted in a significant and selective increase of EEG alpha activity compared to placebo stimulation. These findings suggest the general influence of ABE alpha stimulation on cortical activity. Further studies need to clarify the effects of ABE alpha stimulation on the behavioral and subjective experience level.

In summary, the presented results demonstrated effects of three different ABE application scenarios. These effects were obtained on the physiological level (EEG activity), the behavioral level (response times), and on the level of subjective experience (tiredness self-ratings). Further, effects have been shown to differ significantly from placebo stimulation. At the same time, no negative side effects of the utilized ABE stimulations were observed within the three reported pilot studies (e. g. sickness, headache or an enhancement of drowsiness).

Remarkably, even though the sample sizes of the three reported studies are relatively small, a considerable number of effects were found on a statistically significant level. This indicates effect sizes of at least medium strength. Nevertheless, the next step was to substantiate the reported effects by using a larger sample size.

In the study presented here, the results of the pilot studies are now to be evaluated on a larger sample size. Under similar conditions as in pilot studies 1 and 2, the effectiveness of auditory brain stimulation in the low beta range is to be investigated.

## Material and methods

All procedures performed in studies involving human participants were in accordance with the ethical standards of the institutional and/or national research committee and with the 1964 Helsinki Declaration and its later amendments or comparable ethical standards. The ethics committee of the University of Wuppertal approved the study. Written informed consent was obtained from all individual participants involved in the study. The following section describes the experimental setup and the measurement parameters used. Both subjective sleepiness assessments as well as performance tests and physiological measurements were used.

### Measurements

**EEG responses.** EEG has been recorded from 5 positions (Fz, Cz, C3, C4, Pz position of the 10–20 system as well as A1 and A2 for ground and reference). The mean value of the EEG positions was calculated for evaluation. The EEG recording was done with the Varioport

system (Becker Meditec, [34, 35]), a portable measurement system for the recording of physiological data. EEG Data was preprocessed using highpass (0,2 Hz) and lowpass (50 Hz) filters and recorded at a resolution of 128 Hz.

EEG Activity has been analyzed for 10 minutes time windows:

- Before the stimulation

- During stimulation (minute 1–10)

- During stimulation (minute 11–20)

- During the TAP Task after stimulation

- Driving after stimulation (minute 1–10)

- Driving after stimulation (minute 11–20)

- Driving after stimulation (minute 21–30)

- Driving after stimulation (minute 31–40)

- Driving after stimulation (minute 41–50)

- Driving after stimulation (minute 51–60)

Change in EEG activity has been assessed by means of calculation of difference scores (during/after stimulation–before Stimulation).

**Subjective tiredness ratings: Karolinska Sleepiness Scale (KSS scores).** Since no further insights into subjectively perceived fatigue could be obtained in pilot study 2 by using the higher-resolution 52-point scale, the Karolinska Sleepiness Scale was again used in this study. Since the KSS is a frequently used measurement instrument, this study is also more comparable with similar studies. The Karolinska Sleepiness Scale (KSS, see Table 1) is a Rating Scale for assessing daytime or nighttime sleepiness by rating the alertness/sleepiness on a 9-point Likert-Scale ranging from 1-extremely alert to 9-very sleepy. The subjective rating by the KSS shows a great correlation to reaction times and physiological sleepiness markers like and theta power [36].

The KSS has been conducted in both conditions: (1) Day drive, (2) Night drive. KSS Scores have been assessed before stimulation, 20 Minutes after stimulation, 50 Minutes after stimulation, 80 Minutes after stimulation, 100 Minutes after stimulation (time schedule of the study is shown in Fig 1 below).

Change in subjective experience of tiredness has been assessed by means of calculation of difference scores (after stimulation–before Stimulation).

**Table 1. Items of the Karolinska Sleepiness Scale.**

| Rating | Verbal descriptions |
|---|---|
| 1 | extremely alert |
| 2 | very alert |
| 3 | Alert |
| 4 | fairly alert |
| 5 | neither alert nor sleepy |
| 6 | some signs of sleepiness |
| 7 | sleepy, but no effort to keep alert |
| 8 | sleepy, some effort to keep alert |
| 9 | Very sleepy, great effort to keep alert, fighting sleep |

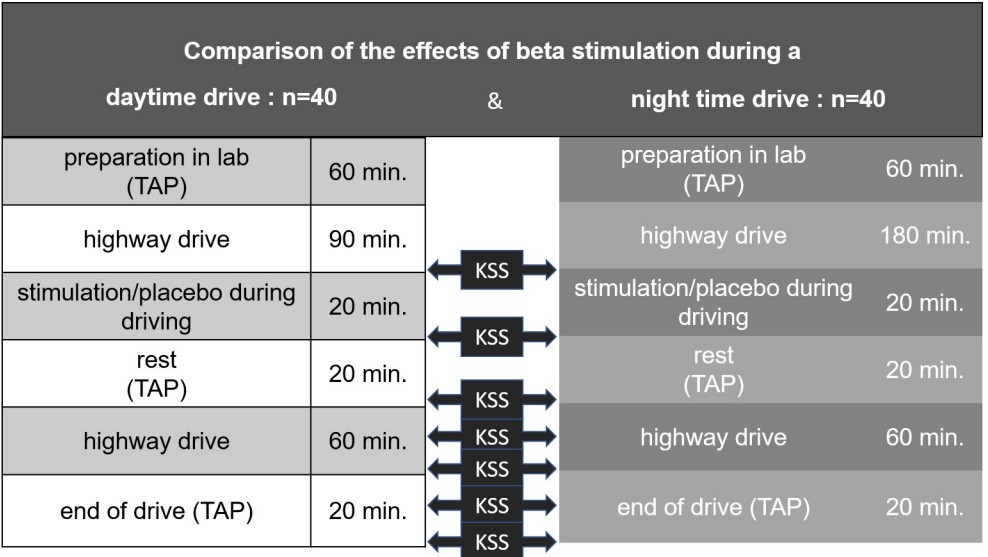

**Fig 1. Study design.**

**Response times (TAP).** The Test of Attentional Performance (German original „Testbatterie zur Aufmerksamkeitsprüfung", TAP, [37]) is a standardized software package, that uses simple reaction paradigms, in which one has to react selectively to well discriminable, non-verbal stimuli by a simple keypress. The performance criteria are the reaction time and any mistakes.

It contains the following subtests: (1) Alertness, (2) Covert Attention Shift, (3) Cross-Modal Integration, (4) Divided Attention, (5) Eye Movement, (6) Flexibility, (7) Go/NoGo, (8) Incompatibility, (9) Sustained Attention, (10) Vigilance, (11) Visual Field/Neglect, (12) Visual Scanning, (13) Working Memory. The development of these tests was based primarily on the needs of neuropsychological diagnostics, which make special demands on the adequate test procedures due to the sometimes high specificity of the deficits as well as the mostly given multiple impairments of the patients. This is reflected particularly in the choice of methods with low complexity by which on the one hand sub-functions are examined and on the other hand the impairments of test performance by sensory and/or motor deficits, memory impairment, speech or other deficiencies are excluded as far as possible. Based on the original Test of Attentional Performance (TAP) the German company "PSYTEST" developed a short form of this test namely TAP-M (Mobility Version). This test was compiled to measure attentional aspects of the ability to drive. The core of these procedures are reaction time tasks of low complexity allowing to evaluate very specific deficiencies. The tasks consist of simple and easily distinguishable stimuli that patients react to by a simple motor response. Thus, the influence of a number of factors that would have an inhibiting effect on testing are kept to a limit. As much as possible, it was attempted to account for factors that may disrupt testing, such as motor problems, visual disorders and language deficits. The subtests in the newly developed test battery TAP-M permit to assess a variety of visuospatial, non-spatial and executive attentional aspects. The TAP-M consists of the newly developed subtests (1) Active Visual Field, (2) Alertness (modified version), (3) Distractibility, (4) Executive Control, (5) Sustained Attention, and the already existing subtests of the TAP (6) Divided Attention, (7) Flexibility, (8) Go/Nogo, (9) Visual Scanning.

For this study, the Covert Attention Shift Test and the Working memory test were used. The reaction tests were conducted before the drive, during a resting phase and after the drive.

Covert shift of attention is the ability to focus visual attention on a section of space without changing the direction of vision. This shift of attention focus can be exogenous through stimuli that catch the eye or endogenous through a deliberate orientation towards an expected or controlled event at a point in space.

In the present task, a central clue (an arrow pointing left or right) is used to indicate the expected side of the target stimulus. The endogenous control of the attentional focus is thus tested. In 80% of the trials the hint is valid, in 20% it is invalid. In the case of an invalid indication, the attention focus is first shifted to the displayed side ("orienting"), when the target stimulus actually appears on the non-displayed side, the focus is shifted again to the target stimulus ("reorienting").

The change in attention was determined by calculating the difference values (after stimulation–before stimulation).

## Experimental design and procedure

The Effects of Beta Stimulation have been compared to Placebo Stimulation during a

- Day Drive (N = 40) and during a

- Night Drive (N = 40) condition.

After arriving at the laboratory, participants received a written information sheet about the procedure and signed an informed consent. Afterwards, a personal information questionnaire was filled in. All participants must have had at least two years of driving experience to take part in the experiment. Drug usage and alcohol consumption within the last 24 hours excluded a person from participation. For safety reasons, each experimental drive was accompanied by a certified driving instructor. Participants were told to inform the experimenter in case they feel unable to continue driving due to severe tiredness. In this case, the driving instructor would drive the car back to the test laboratory. Self-rated tiredness was assessed by means of the Karolinska Sleepiness Scale (KSS). To make participants familiar with the KSS, the experimenter carefully explained the scale and participants completed it a first time with the possibility to ask questions about it. If participants understood how to use the KSS and had no further questions about it, the experimenter started with cleaning and peeling the electrode positions. After this, participants were led to the car, which was parked in the backyard of the laboratory. Electrodes were plugged into the Varioport box, and signals were checked. If necessary, retreatment on the electrode positions took place, to enhance the signal quality. After arrival of the driving instructor the test drive started.

For the highway section, the German "Bundesautobahn 1 (A1)" from Wuppertal to Osnabrück and back was used. To standardize driving speed as much as possible in the real traffic conditions, all participants were instructed to drive at a maximum speed of 120 km/h, which had to be adjusted to predominant speed limits (e.g. at road work sites). 120 km/h were chosen, as this is a common driving speed on the German motorways, which is still below the general recommended maximum speed of 130 km/h. The experimental drive ended at the laboratory. Electrodes were removed and participants received the participation allowance.

**Participants and experimental conditions.** For the purpose of the present analysis, a total sample of N = 80 (47 male, 33 female) participants was assessed. Mean age of the analyzed sample was 38.18 years (SD = 14.1, Minimum: 18, Maximum: 72). Experimental conditions are treatment group which receive ABE in the low beta range (13–21 Hz) and a placebo group which listen to the same music but without the frequency shifts. After the driving period,

**Table 2. Distribution of subjects in experimental conditions.**

|  | A: ABE Beta Stimulation | B: Placebo Music |
| --- | --- | --- |
| 1: Daytime drive | n = 20 | n = 20 |
| 2: Nighttime drive | n = 20 | n = 20 |

participants continued driving on the highway while being exposed to an ABE, that stimulated the higher EEG frequency range (13–21 Hz) by means of a mixture of binaural and monaural beats embedded in a music track for 20 minutes or placebo. After 20 minutes of ABE, the subjects rested at a parking place at the highway. Then, participants continued driving on the highway for another 60 minutes without acoustic stimulation.

The study design was a between-subjects test.

Table 2 shows the subsample characteristics for the experimental conditions.

## Results

The results of the study are presented in the following section. The results are divided into the comparison of the subjective assessment of fatigue, the results of the reaction tests and the examination of the EEG parameters. Multifactor ANOVAs were calculated to determine the effects of the factors placebo vs. beta beats and day vs. night and of interactions. The respective time points of measurement were additionally examined with Welch significance tests, because variance homogeneity was not always given. Cohen's d was calculated as a measure of effect size. Analysis was carried out with the SPSS program package.

### EEG-activity

The cortex activity as assessed by EEG was examined in 10-minute steps.

1. Before the stimulation

2. During stimulation (minute 1–10)

3. During stimulation (minute 11–20)

4. During the TAP Task after 20 min stimulation

5. Driving after stimulation (minute 1–10)

6. Driving after stimulation (minute 11–20)

7. Driving after stimulation (minute 21–30)

8. Driving after stimulation (minute 31–40)

9. Driving after stimulation (minute 41–50)

10. Driving after stimulation (minute 51–60)

Change in EEG activity has been assessed by means of calculation of difference scores (during/after stimulation–before Stimulation).

For each time period, mean values were calculated in the different frequency ranges. The differences between ABE and placebo group as well as for day vs. night condition in the beta band and in the theta band were statistically examined. In the beta band, an increase in activity is expected in the ABE group compared to placebo. In the theta band, activity is expected to decrease in the ABE group compared to the control group.

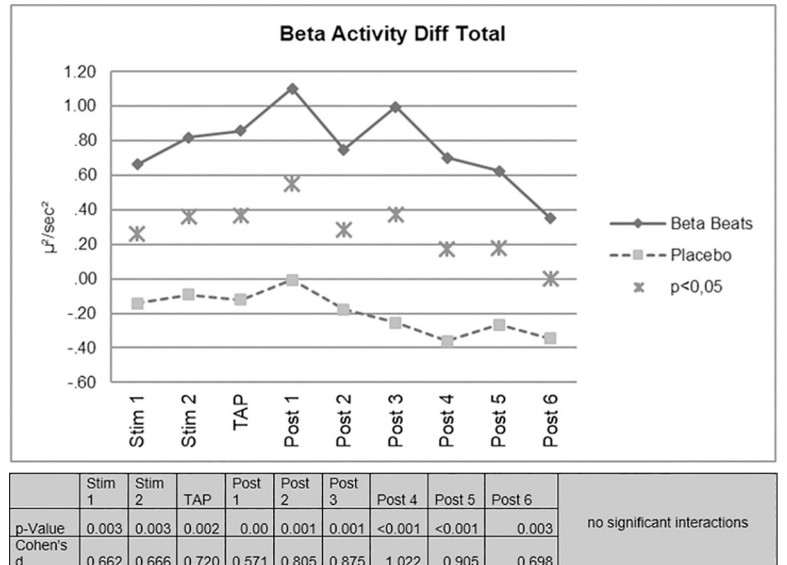

**Fig 2. EEG-beta activity difference scores.**

Beta Stimulation resulted in a significant increase of EEG-beta activity compared to placebo stimulation. The difference remained significant over the duration of the experiment. In each time period investigated, a significant increase in beta activity was observed in the ABE group compared to the control group. The effect was still significant approximately 80 minutes after stimulation (Fig 2).

No significant interactions were found for the day/night factor. On the descriptive level, however, it is noticeable that the effect of increased beta activity in the ABE group appears more pronounced during night driving.

Furthermore, it was examined what percentage of the test persons in the respective group showed an increase in beta activity.

The percentage of subjects showing an increase in EEG-beta activity was higher for Beta Stimulation compared to Placebo (mean of all periods: 76% vs. 41%) (Fig 3).

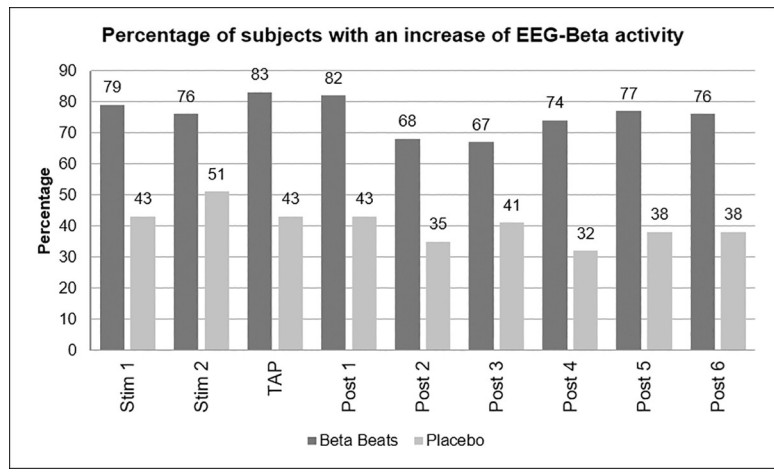

**Fig 3. Percentage of subjects with increase in beta activity.**

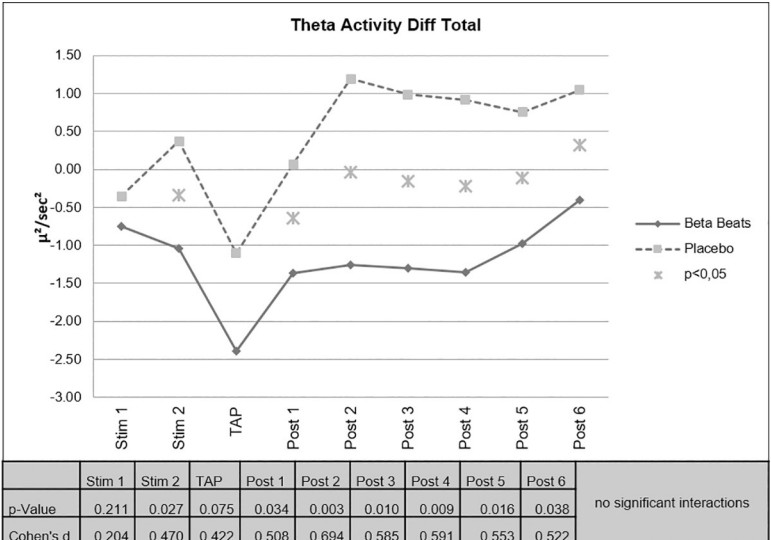

**Fig 4. Changes in EEG theta activity.**

This difference was more pronounced during the Day Drive condition (84% vs. 47%) compared to the Night Drive condition (67% vs. 35%).

Based on the assumption that brain activity in a certain band correlates with a corresponding vigilance state, the change in the beta band was first examined as a correlate to alertness. On the other hand, it is expected that the activity in the theta band, which rather represents fatigue, decreases more in beta beats stimulation than in the placebo condition.

Beta Stimulation resulted in a significant decrease of EEG-theta activity compared to Placebo Stimulation for the periods 2, 3, 4 and 5 after stimulation, i.e. minutes 11–20, 21–30, 31–40 and 41–50 after stimulation. During stimulation (minute 11–20) and after stimulation (minute 1–10 and 51–60) the effect was partly significant. That means, that the effect was still significant approximately 70 minutes after stimulation (Fig 4).

There were no significant interactions of the factor day vs. night. However, on the descriptive level, the difference found in the decrease of theta activity was more pronounced during the Day Drive condition (Fig 5).

## Subjective sleepiness

Fig 6 shows the changes in perceived sleepiness during the experiment for the ABE and the placebo group. Both start at an approximately equal level of sleepiness. It can be shown that subjective experience of tiredness (assessed by means of KSS scores) significantly decreased

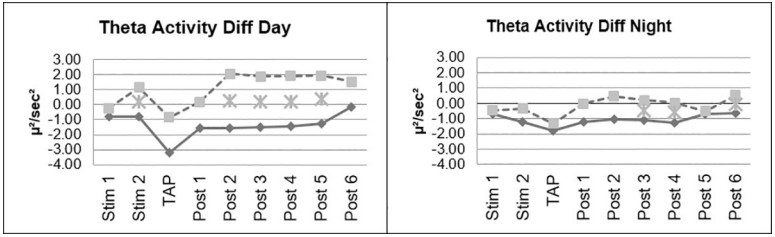

**Fig 5. Changes in EEG theta activity, day vs night.**

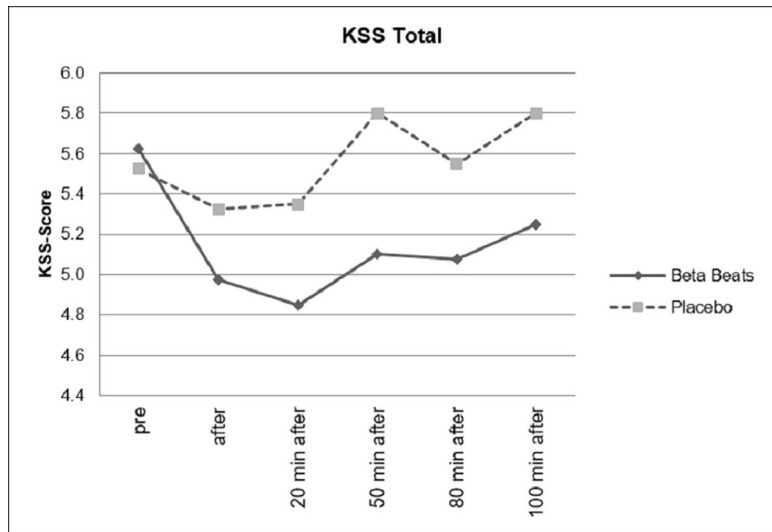

**Fig 6. Changes in subjective sleepiness.**

after Beta Stimulation compared to Placebo Stimulation. The effect was still significant approximately 80 minutes after stimulation.

In Fig 7, the comparison of the difference scores (post-baseline) are shown. Here, even 100 minutes after the stimulation, the effect is still significant. A comparison of the daytime driving group and the night driving group showed no significant interactions.

However, as can be seen in Figs 8 and 9, as expected, the group that drove during the day showed significantly lower levels of fatigue than the group that drove at night.

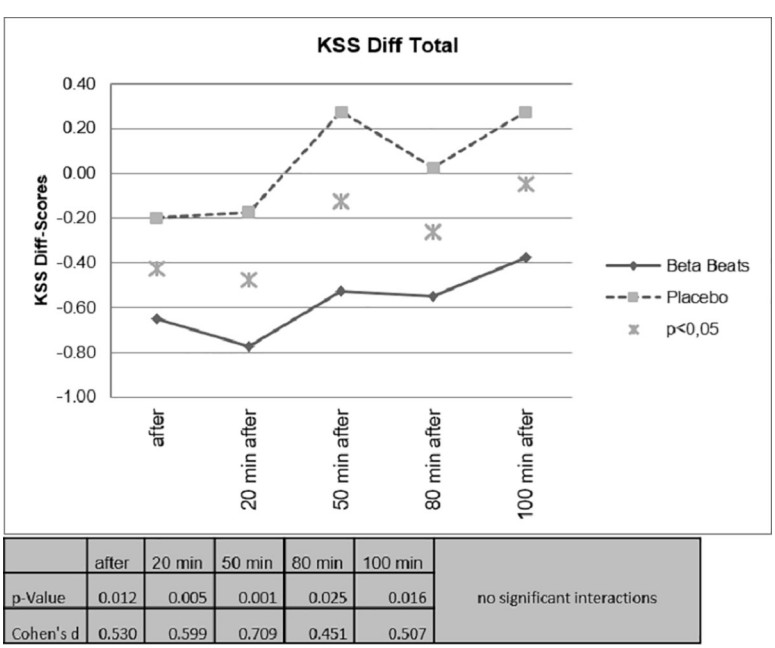

|  | after | 20 min | 50 min | 80 min | 100 min |  |
|---|---|---|---|---|---|---|
| p-Value | 0.012 | 0.005 | 0.001 | 0.025 | 0.016 | no significant interactions |
| Cohen's d | 0.530 | 0.599 | 0.709 | 0.451 | 0.507 |  |

**Fig 7. Difference scores for sleepiness.**

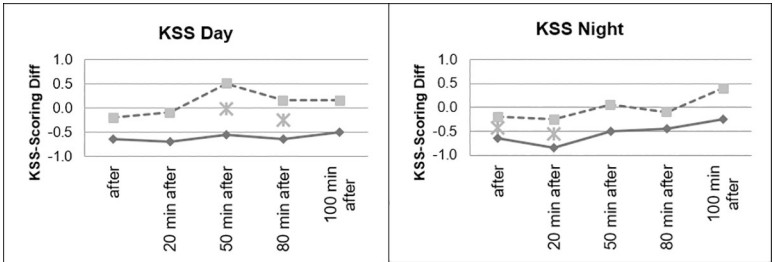

**Fig 8. Sleepiness, divided according to day and night conditions.**

## Response time

Response Times have been assessed with two TAP tasks, the Covert Attention Task and the Working memory Task. The two TAP tasks have been conducted before the start of the drive (Baseline at the laboratory), after stimulation (stop at a resting place), 80 Minutes after stimulation (end of experiment in the parking side) and 100 Minutes after stimulation (back at the laboratory).

Change in Response Times has been assessed by means of calculation of difference scores (after stimulation–before Stimulation).

Faster Response Times could be observed for the Beta Stimulation condition compared to the Placebo condition during the Day Drive and during the Night Drive. This effect was significant for the Covert Attention Task (Fig 10) and significant for the Working Memory Task (Fig 11) right after the stimulation. The effect of faster Response Times in the Covert Attention Tasks for the Beta Stimulation condition was significant still significant 80 minutes after stimulation. No significant interactions could be found for the factor day drive vs. night drive.

Although there were no significant interactions in the day/night factor, some differences could be noticed between the two tasks. The effect on the response time in the Covert attention task was more pronounced during daytime driving, while the groups differed less during nighttime driving (Fig 12).

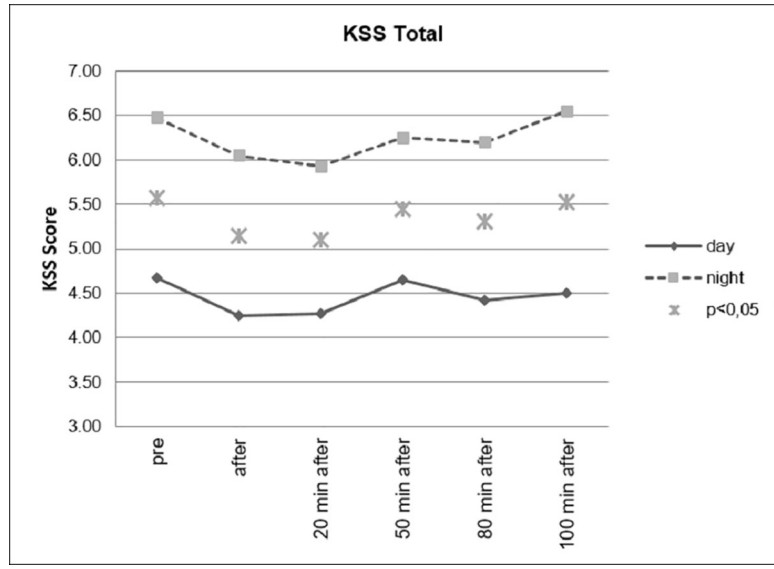

**Fig 9. Sleepiness, day and night comparison.**

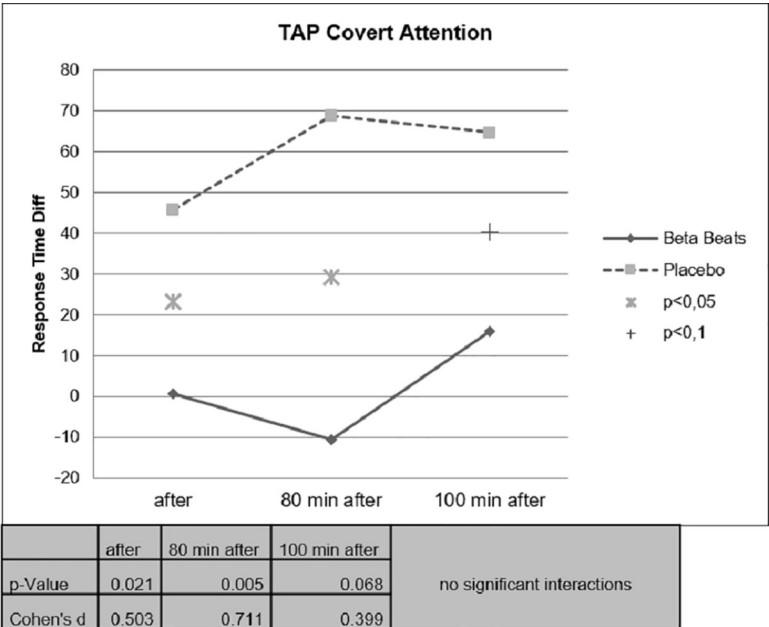

**Fig 10. Difference value covert attention task response time.**

During the working memory task, it could be observed that the ABE group showed stable reaction times during the night drive, while performance of the placebo group continued to decrease in the course of the experiment (Fig 13).

## Discussion

One of the conditions required for a driver to be able to perform the driving task is to be in an optimal state of alertness. This is necessary for the central nervous system to function properly

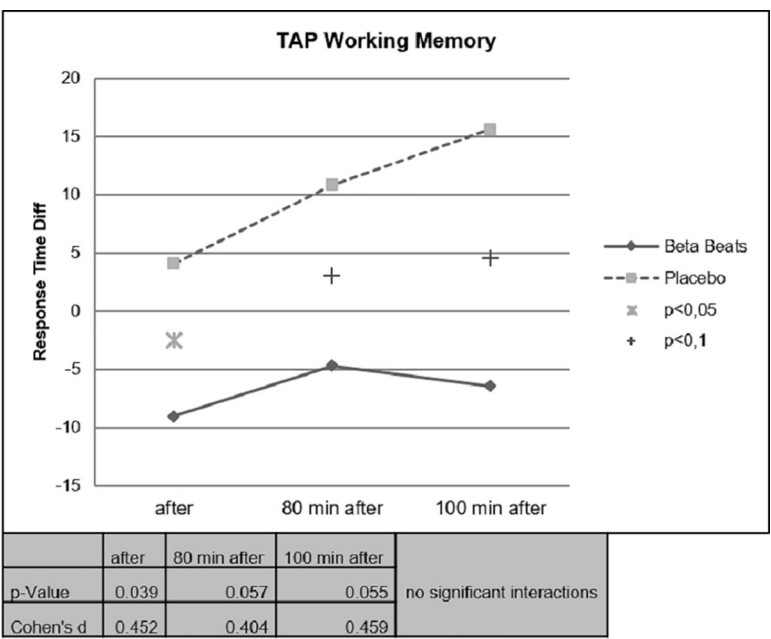

**Fig 11. Working memory task response time.**

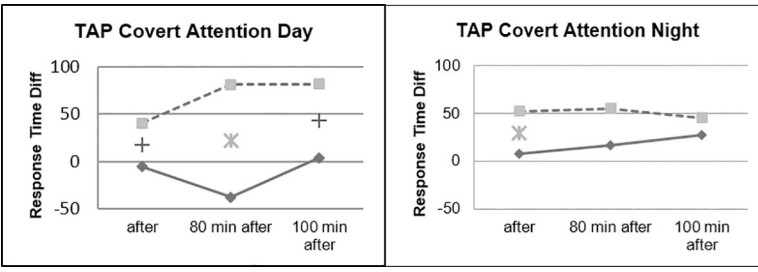

**Fig 12. Covert attention, day vs. night.**

in a given situation. The positive effects of ABE stimulation on the cortical level (measured by beta and theta activity in the EEG) have been demonstrated. Thus, very quickly after stimulation, beta wave power increased and theta wave power decreased in the ABE group. The change in activity in the beta band, which is considered a correlate of vigilance was accompanied by a decrease in theta activity, which in turn is associated with fatigue and inattention. This result shows that the drivers from the ABE group were more awake and therefore their nervous system could provide better cognitive resources. These reactions on the cortical level correspond to the results of the subjective examination (measured by KSS).

The participants in the ABE group reported significantly lower fatigue values over the course of the experiment than the participants in the control group. This started very quickly after the beginning of the stimulation. The stronger the sleepiness is perceived, the less the individual is able to concentrate, and conversely, when the reduction in sleepiness is perceived, motivation for the task may increase. The positive perceptions of the participants are also a guarantee of better acceptability of the proposed countermeasure. A better knowledge of the perception of their states of fatigue and therefore probably a better consideration of these states and the anticipation of their countermeasures. Significant differences were also found at the behavioral level (measured by covert attention test in the TAP), but the working memory test showed less pronounced effects and was partially significant at best.

All of these results, in the three areas that make it possible to characterize the functioning of an organism during the performance of a task: physiological, subjective and behavioral, are consistent for each of the groups respectively and have been demonstrated for day and night conditions. This therefore shows that ABE stimulation is effective both on a state of task-related fatigue and on the circadian effect of fatigue. However, the effects were more frequent and more pronounced in the daytime conditions. Except in the working memory task, where the ABE group showed a constant performance during the duration of the experiment, while the placebo group became slower and slower. This is not a surprising result. The fact that working memory does not improve in the ABE group over time, but also that it deteriorates in

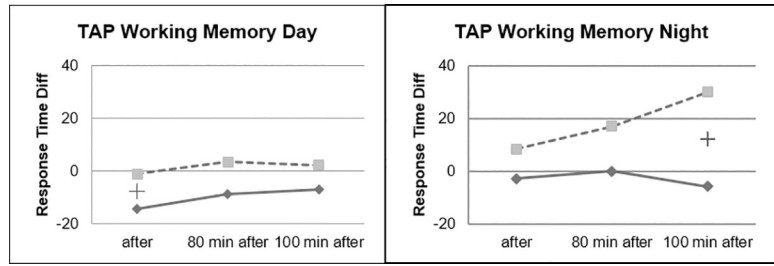

**Fig 13. Working memory task, day vs. night.**

the control group, shows the limits of reactivation in a hypovigilance situation. If we consider the circadian rhythm, whatever the reactivation, the body is less able to react than in a situation of night-time fatigue. Indeed, at night, when alertness is physiologically at its lowest, this is the time when the body recovers, and only sleep allows this recovery. During night driving, the reactivation allowed the ABE group to maintain its working memory capacity, whereas the control group was undergoing this circadian pressure.

No negative side effects of Beta Beats stimulation could be observed. No nausea or other adverse effects have been reported. The central nervous system has the nerve structures to process binaural or monaural stimuli. These stimulations are found in nature and we are continually subjected to them without even being aware of it. The presented intervention simply directs the stimulation towards a desired target, for example, increasing beta power to induce an improvement in arousal. When the power of the stimulation (e.g. 20 Hz in the beta wave band) resonates with the power of the beta waves in a subject, the increase in power will only occur if the beta power is different at that time in the brain. The brain is basically trained to develop more power in the respective frequency range. Our experiences from earlier studies show, that this wave frequency should also not be too far away from the stimulation frequency. It is also noteworthy that the observed effect is constant, i.e. it occurs continuously after the stimulation is applied. It is also surprisingly long-lasting. The stimulation with a length of 20 minutes showed a significant effect in most parameters compared to the placebo for a period up to 100 minutes.

It could be observed that the effect seems to become less clear the more complex the transfer to the behavioral level is: While the subjective assessment of fatigue strongly correlates with the pattern activity objectively recorded in the EEG, this correlation is still present, but less clear, in the more complex Covert Attention Task. In the comparatively complex Working Memory Task of the TAP, effects of Beta Beats stimulation for this task seem to be limited. Nevertheless, we found medium to high effect sizes for most parameters at many measuring times, which also indicates the effectiveness of the approach. There are different levels of performance among people, and some tasks are harder when you are tired than others.

People also have different levels of motivation, which can be exacerbated by fatigue. The explanation for lower significance would be a big difference in cognitive performance and fatigue levels: Motivation and performance are moderated by different levels of fatigue.

The study of tiredness and fatigue in real traffic is an enormous challenge in terms of safety. But especially in the final testing phase of new applications, this type of testing is highly required. This is particularly important in the area of tiredness because the motivation of the participants can influence or moderate the effects of tiredness. In this context motivation is influenced, among other things, by the perception of the risk. Of course, perception of risk differs dramatically between simulator and real driving. The physiological measurements in the present studies could be made continuously throughout the ride, as well as the self-ratings of tiredness. With respect to the behavioral level, due to safety reasons we decided to not conduct a secondary task throughout the ride to record response times, because this could have distracted too much from the driving task. The response times were assessed during driving breaks. Subsequent studies should also weigh very critically the aspect of gaining knowledge against security. Another approach in this context would be the evaluation of the driving behavior data. In real traffic, however, the challenge of standardizing conditions arises because reaction-critical situations cannot be standardized as easily as in a simulator.

Driving speed is also an important external factor that is worth to consider, because it can have an influence on the development of tiredness. Driving fast may be a way to enhance the arousal level. For the presented study, we chose to standardize driving speed at a maximum of 120 km/h, a usual speed on German highways. It should be noted that the average speed driven

has a significant effect on the probability of accidents. For example, a study on the speed limit in Sweden showed that a reduction of the speed limit by 10 km/h led to a reduction of the average speed by 2–3 km/h and saved about 17 lives per year [38]. In a model based on data from Hong Kong, it was calculated that, in addition to the time of day and traffic density, the average speed driven also has an influence on accidents during the day time [39]. More insights could be obtained from a systematic comparison of experimental test drives on different road types with different driving speeds.

## Conclusion

ABE seems to be relevant for the automotive context for several reasons. Tiredness is a major influencing factor for traffic accidents. Nonintrusive countermeasures in this area can play a vital role in enhancing road safety. It seems quite obvious, that user acceptance for ABE applications is given, since listening to music while driving is a very common thing to do, and the technical requirements are given in almost every car. Different ABE applications (stimulation of different EEG frequency ranges) correspond very well with relevant automotive application scenarios (use cases). Reactivation during driving (ABE Beta Stimulation) can be seen in terms of a short-term action. Severe tiredness will not completely disappear with ABE Beta Stimulation, but it may support tired drivers to arrive more safely at the next parking space. During a break regenerative effects can diminish tiredness. Here, ABE could provide a tool to enhance effectiveness of regeneration (ABE Supported Power Nap).

Finally, there is a high interest of the automotive industry in automated detection systems for tiredness currently. An automated detection system for tiredness in combination with an ABE countermeasure seems to be highly user friendly, acceptable and may offer an important contribution to road safety.

In summary, however, the results of this study demonstrated the effectiveness of beta beat stimulation under real road traffic conditions. The effects were shown on the subjective, physiological and performance level. The measure is non-invasive and easy to use in today's vehicles. Based on the results presented here, it can be concluded that Acoustic Brain Entrainment in the sense of Beta Stimulation can be a contribution to traffic safety.

## Supporting information

**S1 File. EEG beta band SPSS file.**
(SAV)

**S2 File. EEG theta band SPSS file.**
(SAV)

**S3 File. KSS SPSS file.**
(SAV)

**S4 File. TAP covert attention SPSS file.**
(SAV)

**S5 File. TAP working memory SPSS file.**
(SAV)

## Author Contributions

**Conceptualization:** Michèle Moessinger.

**Formal analysis:** Ralf Stürmer, Markus Mühlensiep.

**Methodology:** Michèle Moessinger, Ralf Stürmer.

**Project administration:** Michèle Moessinger.

**Writing – original draft:** Michèle Moessinger, Ralf Stürmer, Markus Mühlensiep.

**Writing – review & editing:** Michèle Moessinger, Ralf Stürmer, Markus Mühlensiep.

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
